# Subject-driven Text-to-Image Generation via Apprenticeship Learning

**Wenhu Chen**♠∗    **Hexiang Hu**♠∗    **Yandong Li**♡    **Nataniel Ruiz**♡
**Xuhui Jia**♡    **Ming-Wei Chang**♠    **William W. Cohen**♠

♠Google Deepmind    ♡Google Research

{wenhuchen,hexiang,mingweichang,wcohen}@google.com

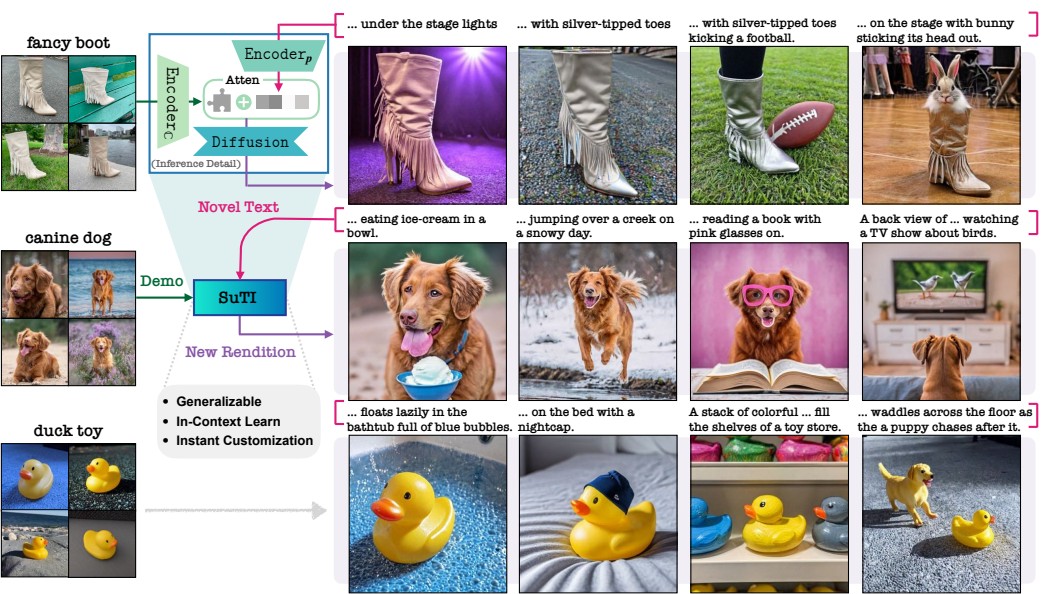

Figure 1: We train *a single* SuTI *model* to generate novel scenes faithfully reflecting given subjects (unseen in training, specified only by 3-5 in-context text→image demonstrations), without any optimization.

## Abstract

Recent text-to-image generation models like DreamBooth have made remarkable progress in generating highly customized images of a target subject, by fine-tuning an "expert model" for a given subject from a few examples. However, this process is expensive, since a new expert model must be learned for each subject. In this paper, we present SuTI, a Subject-driven Text-to-Image generator that replaces subject-specific fine tuning with *in-context* learning. Given a few demonstrations of a new subject, SuTI can instantly generate novel renditions of the subject in different scenes, without any subject-specific optimization. SuTI is powered by *apprenticeship learning*, where a single apprentice model is learned from data generated by a massive number of subject-specific expert models. Specifically, we mine millions of image clusters from the Internet, each centered around a specific visual subject. We adopt these clusters to train a massive number of expert models, each specializing in a different subject. The apprentice SuTI model then learns to imitate the behavior of these fine-tuned experts. SuTI can generate high-quality and customized subject-specific images 20x faster than optimization-based SoTA methods. On the challenging DreamBench and DreamBench-v2, human evaluation shows that SuTI significantly outperforms other existing models.

---

∗Core Contribution

37th Conference on Neural Information Processing Systems (NeurIPS 2023).

# 1 Introduction

Recent text-to-image generation models [1] have shown great progress in generating highly realistic, accurate, and diverse images from a given text prompt. These models are pre-trained on web-crawled image-text pairs like LAION [2] with autoregressive backend models [3, 4] or diffusion backend models [5, 1]. Though achieving unprecedented success in generating highly accurate images, these models are not able to customize to a given subject, like a specific dog, shoe, backpack, etc. Therefore, *subject-driven text-to-image generation*, the task of generating highly customized images with respect to a target subject, has attracted significant attention from the community. Subject-driven image generation is related to text-driven image editing but often needs to perform more sophisticated transformations to source images (e.g., rotating the view, zooming in/out, changing the pose of subject, etc.) so existing image editing methods are generally not suitable for this new task.

Current subject-driven text-to-image generation approaches are slow and expensive. While different approaches like DreamBooth [6], Imagic [7], and Textual Inversion [8] have been proposed, they all require fine-tuning specific models for a given subject on one or a few demonstrated examples, which typically takes at least 10-20 minutes[2] to specialize the text-to-image model checkpoint for the given subjects. These approaches are time-consuming as they require back-propagating gradients over the entire model for hundreds or even thousands of steps per customization. Moreover, they are space-consuming as they require storing a subject-specific checkpoint per subject. To avoid the excessive cost, Re-Imagen [9] proposed a retrieval-augmented text-to-image framework to train a subject-driven generation model in a weakly-supervised fashion. Since the retrieved neighbor images are not guaranteed to contain the same subjects, the model does not perform as good as DreamBooth [6] for the task of subject-driven image generation.

To avoid excessive computation and memory costs, we propose to train a single subject-driven text-to-image generation model that can perform on-the-fly subject customization. Our method is dubbed Subject-driven Text-to-Image generator (SuTI), which is trained with a novel *apprenticeship learning* algorithm. Unlike standard apprenticeship learning which only focuses on learning from one expert, our apprentice model imitates the behaviors of a massive number of specialized expert models. After such training, SuTI can instantly adapt to unseen subjects and unseen or even compositional descriptions with only 3-5 in-context demonstrations within 30 seconds (on a Cloud TPU v4).

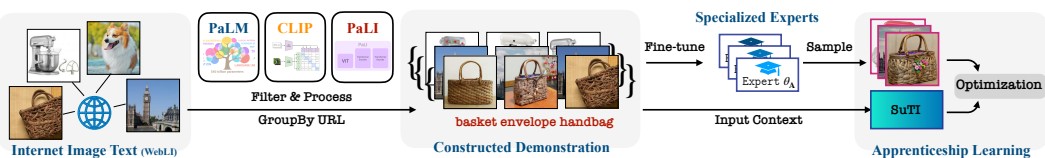

Figure 2: Conceptual Diagram of the Learning Pipeline

Figure 2 presents a conceptual diagram of the learning and data preparation pipeline. We first group the images in WebLI [10] by their source URL to form tiny image clusters because images from the same URL are likely to contain the same subject. We then performed extensive image-to-image and image-to-text similarity filtering to retain image clusters that contain highly similar content. For each subject image cluster, we fine-tuned an expert model to specialize in the given subject. Then, we use the fine-tuned experts to synthesize new images given unseen creative captions proposed by large language models. However, the tuned expert models are not perfect and prone to errors, therefore, we adopt a quality validation metric to filter out a large portion of degraded outputs. The remaining high-quality images are provided as a training signal to teach the apprentice model SuTI to perform subject-driven image generation with high fidelity. During inference, the trained SuTI can attend to a few in-context demonstrations to synthesize new images on the fly.

We evaluate SuTI on various tasks such as subject re-contextualization, attribute editing, artistic style transfer, and accessorization. We compare SuTI with existing models on DreamBench [6], which contains diverse subjects from wide categories accompanied by some prompt templates. We compute the CLIP-I/CLIP-T and DINO scores of SuTI's generated images on this dataset and compare them with DreamBooth. The results indicate that SuTI can outperform DreamBooth while having 20x faster inference speed and significantly less memory footprint.

---

[2]Running on A100 according to public colab: `https://huggingface.co/sd-dreambooth-library` and `https://huggingface.co/docs/diffusers/training/text_inversion`.

Further, we manually created 220 diverse and compositional prompts regarding the subjects in Dream-Bench for human evaluation, which is dubbed the DreamBench-v2 dataset. We then comprehensively compare with other baselines like InstructPix2Pix [11], Null-Text Inversion [12], Imagic [7], Textual Inversion [8], Re-Imagen [9], and DreamBooth [6] on DreamBench-v2. Our human evaluation results indicate that SuTI is 5% higher than DreamBooth and at least 30% better than the other baseline in terms of human evaluation. We conduct detailed fine-grained analysis and found that SuTI's textual alignment is significantly better than DreamBoothm while its subject alignment is slightly better than DreamBooth. However, DreamBooth's outputs are still better in the photorealism aspect, especially in terms of fine-grained detail preservation.

We summarize our contributions in the following aspects:

- We introduce the SuTI model, a subject-driven text-to-image generator that performs instant and customized generation for a visual subject with few (image, text) exemplars, *all in context*.
- We employ the *apprenticeship learning* to train one single apprentice SuTI model to imitate half a million fine-tuned subject-specific experts on a large-scale seed dataset, leading to a generator model that generalizes to unseen subjects and unseen compositional descriptions.
- We perform a comprehensive set of automatic and human evaluations to show the capability of our model on generating highly faithful and creative images on DreamBench and DreamBench-v2.

To facilitate the reproducibility of our model performance, we release the SuTI model API as a Google Cloud Vertex AI model service, under the production name 'Instant tuning'[3].

## 2   Preliminary

In this section, we introduce the key concepts and notations about subject-driven image-text data, then discuss the basics of text-to-image diffusion models.

**Diffusion Models.** Diffusion models [13] are latent variable models, parameterized by $\Theta$, in the form of $p_\Theta(\boldsymbol{x}_0) := \int p_\Theta(\boldsymbol{x}_{0:T})d\boldsymbol{x}_{1:T}$, where $\boldsymbol{x}_1, \cdots, \boldsymbol{x}_T$ are "noised" latent versions of the input image $\boldsymbol{x}_0 \sim q(\boldsymbol{x}_0)$. Note that the dimensionality of both latents and the image is the same throughout the entire process, with $\boldsymbol{x}_{0:T} \in \mathbb{R}^d$ and $d$ equals the product of <height, width, # of channels>. The process that computes the posterior distribution $q(\boldsymbol{x}_{1:T}|\boldsymbol{x}_0)$ is also called the forward (or diffusion) process, and is implemented as a predefined Markov chain that gradually adds Gaussian noise to the data according to a schedule $\beta_t$:

$$q(\boldsymbol{x}_{1:T}|\boldsymbol{x}_0) = \prod_{t=1}^T q(\boldsymbol{x}_t|\boldsymbol{x}_{t-1}); \quad q(\boldsymbol{x}_t|\boldsymbol{x}_{t-1}) := \mathcal{N}(\boldsymbol{x}_t; \sqrt{1-\beta_t}\boldsymbol{x}_{t-1}, \beta_t\boldsymbol{I}) \tag{1}$$

Diffusion models are trained to learn the image distribution by reversing the diffusion Markov chain. Theoretically, this reduces to learning to denoise $\boldsymbol{x}_t \sim q(\boldsymbol{x}_t|\boldsymbol{x}_0)$ into $\boldsymbol{x}_0$, with a time re-weighted square error loss—see [14] for the complete proof:

$$\mathbb{E}_{(\boldsymbol{x}_0, \boldsymbol{c}) \sim D}\{\mathbb{E}_{\boldsymbol{\epsilon}, t}[w_t \cdot ||\hat{\boldsymbol{x}}_\theta(\boldsymbol{x}_t, \boldsymbol{c}) - \boldsymbol{x}_0||_2^2]\} \tag{2}$$

where $D$ is the training dataset containing (image, condition) = $(\boldsymbol{x}_0, \boldsymbol{c})$ pairs, the condition normally refers to the input text prompt. In practice, $w_t$ can be simplified as 1 according to [14, 15].

**Subject-Driven Text-to-Image Generation.** Existing subject-driven generation models [6, 16, 8] often fine-tune a pre-trained text-to-image diffusion model on a set of provided demonstrations $\mathbb{C}_s$ about a specific subject $s$. Formally, such demonstration contains a set of text and image pairs $\mathbb{C}_s = \{(\boldsymbol{x}_k, \boldsymbol{c}_k)\}_k^{K_s}$, centered around the subject $s$. Images $\boldsymbol{x}_k$ contains images of the same subject $s$, while $\boldsymbol{c}_s$ is a short description of images $\boldsymbol{x}_k$. DreamBooth [6] also requires an additional $\bar{\mathbb{C}}_s$, which contains images about different subjects of the same category as $s$ for prior preservation. To obtain a customized diffusion model $\hat{\boldsymbol{x}}_{\theta_s}(\boldsymbol{x}_t, \boldsymbol{c})$, we need to optimize the following loss function:

$$\theta_s = \arg\min_\theta \mathbb{E}_{(\boldsymbol{x}_0, \boldsymbol{c}) \sim \mathbb{C}_s \cup \bar{\mathbb{C}}_s}\{\mathbb{E}_{\boldsymbol{\epsilon}, t}[||\hat{\boldsymbol{x}}_\theta(\boldsymbol{x}_t, \boldsymbol{c}) - \boldsymbol{x}_0||_2^2]\} \tag{3}$$

The customized diffusion model $\hat{\boldsymbol{x}}_{\theta_s}(\boldsymbol{x}_t, \boldsymbol{c})$ has shown impressive capabilities to generate highly faithful images of the specified subject $s$.

---

[3]Generally available at https://cloud.google.com/vertex-ai/docs/generative-ai/image/fine-tune-model

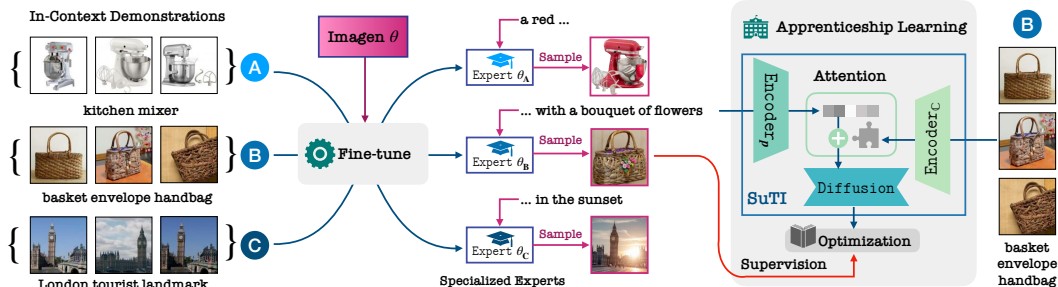

Figure 3: Overview of the apprentice learning pipeline for `SuTI`. Left part shows the customization procedure for expert models, and the right parts shows the `SuTI` model that imitates the behaviors of differently customized experts. Note that this framework can cope with expert models of *arbitary architecture and model family*.

# 3 Apprenticeship Learning from Subject-specific Experts

**Notation.** Figure 3 presents the concrete workflow of learning. Our method follows apprenticeship learning [17] with two major component, *i.e.*, the expert diffusion models $\hat{\boldsymbol{x}}_{\theta_s}(\boldsymbol{x}_t, \boldsymbol{c})$ parameterized by $\theta_s$ regarding subject $s \in \mathbb{S}$ and apprentice diffusion model $\hat{\boldsymbol{x}}_{\Theta}(\boldsymbol{x}_t, \boldsymbol{c}, \mathbb{C}_s)$ parameterized by $\Theta$. The apprentice model takes an additional set of image-text demonstrations $\mathbb{C}_s$ as input. We use $\mathbb{S}$ to denote the superset of subjects we include in the training set.

**Dataset.** The training set $\mathcal{D}_{\mathbb{S}}$ contains a collection of $\{\mathbb{C}_s, \boldsymbol{p}_s\}_{s \in \mathbb{S}}$, where each entry contains an image-text cluster $\mathbb{C}_s$ accompanied by an unseen prompt $\boldsymbol{p}_s$. The image-text cluster $\mathbb{C}_s$ contains a set of 3-10 image-text pairs. The unseen prompt is an imaginary caption proposed by PaLM [18]. For example, if $\boldsymbol{c}$ is 'a photo of berry bowl', then $\boldsymbol{p}_s$ would be an imaginary caption like 'a photo of berry bowl floating on the river'. We describe the dataset construction process to section 4.

**Learning.** To obtain an expert $\hat{\boldsymbol{x}}_{\theta_s}(\boldsymbol{x}_t, \boldsymbol{c})$ on a subject $s$, we fine-tune a pre-trained diffusion model [1] on the image cluster $\mathbb{C}_s$ with the denoising loss as:

$$\theta_s = \arg\min_{\theta} \mathbb{E}_{(\boldsymbol{x}_s, \boldsymbol{c}) \sim \mathbb{C}_s} \{ \mathbb{E}_{\boldsymbol{\epsilon}, t} [||\hat{\boldsymbol{x}}_{\theta}(\boldsymbol{x}_t, \boldsymbol{c}) - \boldsymbol{x}_s||_2^2] \} \quad (4)$$

where $\boldsymbol{x}_t \sim q(\boldsymbol{x}_t | \boldsymbol{x}_s)$. The training is similar to Eqn. 3 except that we do not have negative examples for prior preservation because finding the negative examples from the same class is expensive.

Once an expert model is trained, we use it to sample images $\boldsymbol{y}_s$ for the unseen text description $\boldsymbol{p}_s$ to guide the apprentice `SuTI` model. We gather the outputs from the massive amount of expert models and then use CLIP filtering to construct a dataset $G$. Similarly, we fine-tune the apprentice model $\hat{\boldsymbol{x}}_{\Theta}(\boldsymbol{x}_t, \boldsymbol{p}_s, \mathbb{C}_s)$ with the denoising loss on the pseudo target generated by the expert:

$$\Theta = \arg\min_{\Theta} \mathbb{E}_{(\boldsymbol{y}_s, \boldsymbol{p}_s, \mathbb{C}_s) \sim G} \{ \mathbb{E}_{\boldsymbol{\epsilon}, t} [||\hat{\boldsymbol{x}}_{\Theta}(\boldsymbol{x}_t, \boldsymbol{p}_s, \mathbb{C}_s) - \boldsymbol{y}_s||_2^2] \} \quad (5)$$

where $\boldsymbol{x}_t \sim q(\boldsymbol{x}_t | \boldsymbol{y}_s)$, and the training triples $(\boldsymbol{y}_s, \boldsymbol{p}_s, \mathbb{C}_s)$ are drawn from $G$.

**Algorithm.** We formally introduce our learning algorithm in the Algorithm 1. To improve the training efficiency, we use distributed training algorithm to accelerate the training process. At each training step, we randomly sample a batch $\{B_{s_i}\}_{i=1}^{\mathtt{K}}$ of size $\mathtt{K}$ from the dataset $\mathcal{D}_{\mathbb{S}}$, with $B_{s_i} = (\mathbb{C}_{s_i}, \boldsymbol{p}_{s_i})$. We then fine-tune $\mathtt{K}$ expert models separately w.r.t. Eqn. 4 in parallel, across $\mathtt{K}$ different TPU cores. For every subject $s$ inside the batch $B_s$, we use the corresponding expert model $\theta_s$ to synthesize the image $\boldsymbol{y}_s$ given the unseen prompt $\boldsymbol{p}_s$. As not all expert models can generate highly faithful images, we introduce a quality assurance step to validate the synthesized images. Particularly, we measure the quality of an expert's generation by the delta CLIP score [19] $\Delta(\boldsymbol{y}_s, \mathbb{C}_s, \boldsymbol{p}_s)$, which is used to decide whether a sample should be included in the dataset $G$. This ensures the high quality of the text-to-image training signal for `SuTI`. Specifically, the delta CLIP score is computed as the increment of CLIP score of $\boldsymbol{y}_s$ over the demonstrated images $\boldsymbol{x} \in \mathbb{C}_s$:

$$\Delta(\boldsymbol{y}_s, \mathbb{C}_s, \boldsymbol{p}_s) = \text{CLIP}(\boldsymbol{y}_s, \boldsymbol{p}_s) - \max_{\boldsymbol{x} \in \mathbb{C}_s} \text{CLIP}(\boldsymbol{x}, \boldsymbol{p}_s) \quad (6)$$

We then feed $G$ as a training batch to update the parameter $\Theta$ of the apprentice model using Eqn. 5. In all our experiments, we set $\mathtt{K} = 400$, with each TPU core training an expert model.

**Inference.** To perform subject-driven text-to-image generation, the trained `SuTI` takes 3-5 image-text pairs as the demonstration to generate new images based on the given text description. No

---

**Algorithm 1** Apprenticeship Learning from a Large Crowd of Specialized Expert Models

---

1: **Input:** Dataset $\mathcal{D}_{\mathbb{S}} = \{(\mathbb{C}_s, \boldsymbol{p}_s)\}_{s \in \mathbb{S}}$ containing subject image cluster $\mathbb{C}_s$ and unseen prompt $\boldsymbol{p}_s$
2: **Input:** Pre-trained diffusion model parameterized by $\theta$
3: **Output:** Apprentice diffusion model parameterized by $\Theta$
4: Initialize a buffer $G = \varnothing$
5: **while** $\mathcal{D}_{\mathbb{S}} \neq \varnothing$ **do**
6:     $\{B_{s_i}\}_{i=1}^{\mathtt{K}} = \mathtt{Dequeue}(\mathcal{D}_{\mathbb{S}}, \mathtt{K})$, where $B_{s_i} = (\mathbb{C}_{s_i}, \boldsymbol{p}_{s_i})$
7:     Fine-tune $\mathtt{K}$ expert models $\theta_{s_1}, \ldots, \theta_{s_\mathtt{K}}$ on $\{B_{s_i}\}_{i=1}^{\mathtt{K}}$ in parallel, based on Eqn. 4
8:     **for** $i = 1$ **to** $\mathtt{K}$ **do**
9:         Sample a subject-specific generation $\boldsymbol{y}_{s_i}$ with DDPM using $\hat{\boldsymbol{x}}_{\theta_{s_i}}(\boldsymbol{x}_t, \boldsymbol{p}_{s_i})$
10:         **if** $\Delta(\boldsymbol{y}_{s_i}, \mathbb{C}_{s_i}, \boldsymbol{p}_{s_i}) > \lambda$ **then**
11:            $G = \mathtt{Enqueue}(G, (\boldsymbol{y}_s, \mathbb{C}_{s_i}, \boldsymbol{p}_{s_i}))$
12:         **end if**
13:     **end for**
14: **end while**
15: Train $\hat{\boldsymbol{x}}_{\Theta}$ on the generated demonstration $G$, based on the Eqn. 5

---

optimization is needed during inference time. The only overhead of `SuTI` is the cost of encoding these 3-5 image-text pairs and the attention computation, which is more affordable. Our inference speed is roughly in the same order as the original text-to-image generator [1].

## 4   Mining and Generating Subject-driven Text-to-Image Demonstrations

In this section, we discuss how we created the seed dataset $\mathcal{D}_{\mathbb{S}}$ by mining images and text over the Internet. We construct the seed dataset using the WebLI [10, 20] dataset. Particularly, we derive our initial image cluster via subsampling the Episodic WebLI data [20], which grouped Web images from the same URL. Then we filter the clusters to ensure high intra-cluster visual similarity, using image matching models. The filtered set of image-text clusters is then denoted $\{\mathbb{C}_s\}_{s \in \mathbb{S}}$.

After obtaining the subject-driven image clusters, we further prompt a large language model [18] to generate a description about the subject, with the goal of creating descriptions of plausible imaginary visual scenes. The generating instances of the descriptions will require skills like *subject re-contextualization*, *attribute editing*, *artistic style transfer*, and *accessorization*. We denote the generated unseen captions as $p_s$. Together with $\mathbb{C}_s$, this forms the final dataset $\mathcal{D}_{\mathbb{S}}$. More details regarding the grouping and filtering are provided in the Appendix.

The dataset $\mathcal{D}_{\mathbb{S}}$ contains a total of 2M $(\mathbb{C}_s, p_s)$ pairs. Using the aforementioned delta CLIP score filtering (using a high threshold $\lambda = 0.02$), we remove low-quality synthesized images $\boldsymbol{y}_s$ from the expert model, finally obtaining a dataset $G$ with $\sim$500K $(\mathbb{C}_s, p_s)$ effective training pairs for the following apprenticeship learning.

## 5   Experiment

In this paper, we only train `SuTI` on the text $\rightarrow$ 64x64 diffusion model and retain the original 256x256 and 1024x1024 super-resolution as it is from Imagen [1].

**Expert Models.** The expert model is initialized from the original 2.1B Imagen 64x64 model. We tune each model on a single TPU core (32 GB) for 500 steps using Adafactor optimizer with a learning rate of 1e-5, which only takes 5 minutes to finish. We use classifier-free guidance to sample new images, where the guidance weight is set to 30. To avoid excessive memory costs, we use fine-tuned experts to sample pseudo-target images and then write the samples as separate files. `SuTI` will read these files asynchronously to maximize the training speed. Our expert models have a few distinctions from the DreamBooth [6]: 1) we adopt Adafactor instead of Adam optimizer, 2) we do not include any class word token like '[DOG] dog' in the prompt. 3) we do not include in-class negatives for prior preservation. Though our expert model is weaker than DreamBooth, such design choices significantly reduce time/space costs to enable us to train millions of experts with reasonable resources.

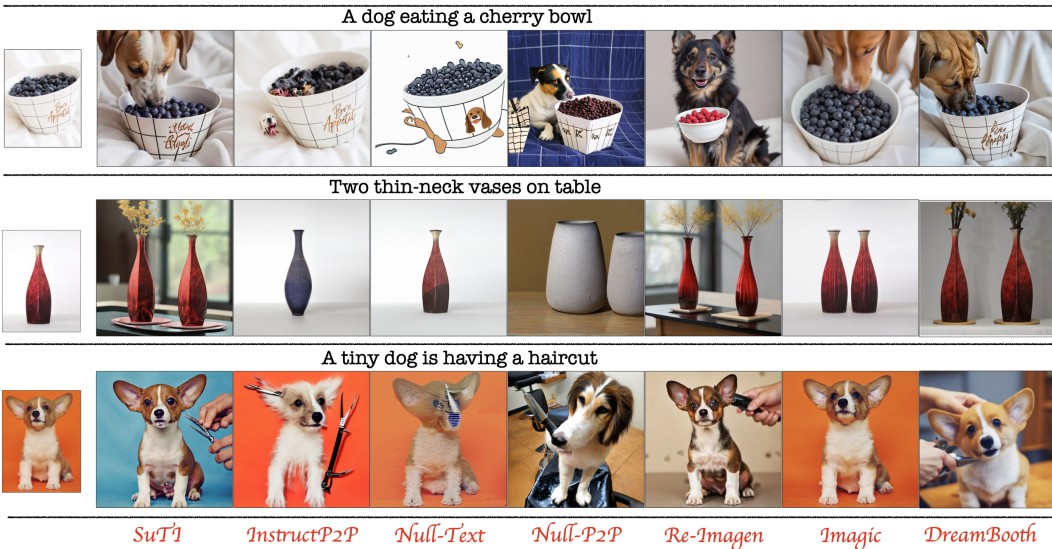

Figure 4: Comparison with other Image Editing and Image Personalization Models.

**Apprentice Model.** The apprentice model contains 2.5B parameters, which is 400M parameters larger than the original 2.1B Imagen 64x64 model. The added parameters are coming from the extra attention layers over the demonstrated image-text inputs. We adopt the same architecture as Re-Imagen [9], where the additional image-text pairs are encoded by re-using the UNet DownStack, and the attention layers are added to the down UNet DownStack and UpStack at different resolutions.

We initialize our model from Imagen's checkpoint. For the additional attention layers, we use random initialization. The apprentice training is performed on 128 Cloud TPU v4 chips. We train the model for a total of 150K steps. We use an Adafactor optimizer with a learning rate of 1e-4. We use 3 demonstrations during training, while the model can generalize to leverage any number of demonstrations during inference. We show our ablation studies in the following section.

**Inference.** We normally provide 4 demonstration image-text pairs to SuTI during inference. Increasing the number of demonstrations does not improve the generation quality much. We use a lower classifier-free guidance weight of 15 with DDPM [14] sampling strategy.

## 5.1 Datasets and Metrics

**DreamBench.** In this paper, we use the DreamBench dataset proposed by DreamBooth [6]. The dataset contains 30 subjects like backpacks, stuffed animals, dogs, cats, clocks, etc. These images are downloaded from Unsplash. The original dataset contains 25 prompt templates covering different skills like recontextualization, property modification, accessorization, etc. In total, there are a total of 750 unique prompts generated by the template. We follow the original paper to generate 4 images for each prompt to form the 3000 images for robust evaluation. We follow DreamBooth to adopt DINO, CLIP-I to evaluate the subject fidelity, and CLIP-T to evaluate the text fidelity.

**DreamBench-v2.** To further increase the difficulty and diversity of DreamBench, we annotate 220 prompts for the 30 subjects in DreamBench as DreamBench-v2. We gradually increase the compositional levels of the prompt to increase the difficulty, like 'back view of [dog]' → 'back view of [dog] watching TV' → 'back view of [dog] watching TV about birds'. This enables us to perform a breakdown analysis to understand the model's compositional capabilities.

We use human evaluation to measure the generation quality in DreamBench-v2. Specifically, we aim at measuring the following three aspects: (1) the subject fidelity score $s_s$ measures whether the subject is being preserved, (2) the textual fidelity score $s_t$ measures whether it is aligned with the text description, (3) the photorealism score $s_p$ measures whether the image contains artifacts or blurry subjects. These are all binary scores, which are averaged over the entire dataset. We combine them as an overall score $s_o = s_s \wedge s_t \wedge s_p$, which is the most stringent score.

## 5.2 Main Results

**Baselines.** We provide a comprehensive list of baselines to compare with the proposed `SuTI` model:

- *DreamBooth* [6]: a fine-tuning method with space consumption is $|M| \times |\mathbb{S}|$, where $|M|$ is the model size and $|\mathbb{S}|$ is the number of subjects.

- *Textual Inversion* [8]: a fine-tuning method with space consumption is $|E| \times |\mathbb{S}|$ with the embedding size of $|E|$, note that $|E| \ll |M|$.

- *Null-Text Inversion* [12]: a fine-tuning method with space consumption is $|T| \times |E| \times |\mathbb{S}|$.

- *Imagic* [7]: a fine-tuning-based method with largest space consumption among all models as it requires training $|M| \times |\mathbb{S}| \times |\mathbb{P}|$, where $|\mathbb{P}|$ is the number of a text prompt $\mathbb{P} = \{\boldsymbol{p}_s\}$ for the subject set $\mathbb{S}$.

- *InstructPix2Pix* [11]: a non-tuning method, which can generate and edit a given image really fast within a few seconds. There is no additional space consumption.

- *Re-Imagen* [9]: a non-tuning method, which will take a few images as input and then attend to those retrievals to generate a new image. There is no additional space consumption.

**Experimental Results.** We show our automatic evaluation results on the DreamBench in Table 1. We can observe that `SuTI` can perform better or on par with DreamBooth on all of the metrics. Specifically, `SuTI` outperforms DreamBooth on the DINO score by 5%, which indicates that our method is better at preserving the subject's visual appearance. In terms of the CLIP-T score, our method is almost the same as DreamBooth, indicating an equivalent capability in terms of textual alignment. These results indicate that `SuTI` has achieved promising generalization to a wide variety of visual subjects, without being trained on the exact instances.

| Methods | Backbone | DINO ↑ | CLIP-I ↑ | CLIP-T ↑ |
|---|---|---|---|---|
| Real Image (Oracle) | - | 0.774 | 0.885 | - |
| DreamBooth [6] | Imagen [1] | 0.696 | 0.812 | **0.306** |
| DreamBooth [6] | SD [21] | 0.668 | 0.803 | 0.305 |
| Textual Inversion [8] | SD [21] | 0.569 | 0.780 | 0.255 |
| Re-Imagen [9] | Imagen [1] | 0.600 | 0.740 | 0.270 |
| Ours: SuTI | Imagen [1] | **0.741** | **0.819** | 0.304 |

Table 1: Automatic Evaluation on the DreamBench.

We further show our human evaluation results on the DreamBench-V2 in Table 2. It shows the related rankings for the additional storage cost and reported the average inference time measure for inferring on each subject. As can be seen, `SuTI` is able to outperform DreamBooth by 5% on the overall score mainly due to much higher textual alignment. In contrast, all the other existing baselines are getting much lower human evaluation score (< 42%).

**Comparisons.** We compare our generation results with other methods in Figure 4. As can be seen, `SuTI` can generate images highly faithful to the demonstrated subjects. Though `SuTI` is still

| Methods | Backbone | Space | Time | Subject ↑ | Text ↑ | Photorealism ↑ | Overall ↑ |
|---|---|---|---|---|---|---|---|
| Models requiring test-time tuning | | | | | | | |
| Textual Inversion [8] | SD [21] | $ | 30 mins | 0.22 | 0.64 | 0.90 | 0.14 |
| Null-Text Inversion [12] | Imagen [1] | $$ | 5 mins | 0.20 | 0.46 | 0.70 | 0.10 |
| Imagic [7] | Imagen [1] | $$$$ | 70 mins | 0.78 | 0.34 | 0.68 | 0.28 |
| DreamBooth [6] | SD [21] | $$$ | 6 mins | 0.74 | 0.53 | 0.85 | 0.47 |
| DreamBooth [6] | Imagen [1] | $$$ | 10 mins | 0.88 | 0.82 | **0.98** | 0.77 |
| InstructPix2Pix [11] | SD [21] | - | 10 secs | 0.14 | 0.46 | 0.42 | 0.10 |
| Re-Imagen [9] | Imagen [1] | - | 20 secs | 0.70 | 0.65 | 0.64 | 0.42 |
| Ours: SuTI | Imagen [1] | - | 20 secs | **0.90** | **0.90** | 0.92 | **0.82** |

Table 2: Human Evaluation on the DreamBench-v2. We report an approximated average inference time (averaged over subjects) and the relative rankings of the space cost (more $: more expensive). Methods that do not fine-tune in test-time requires no additional storage (denoted by -).

missing some local textual (words on the bowl gets blurred) or colorization (dog hair color gets darker), the nuance is almost unperceivable for humans. The other baselines like InstructPix2Pix [11], and Null-Text Inversion [12] are not able to perform very sophisticated transformations. Textual Inversion [8] cannot achieve satisfactory results even with 30 minutes of tuning. Re-Imagen [9] though gives reasonable outputs, the subject preservation is much weaker than `SuTI`. Imagic [7] also generates reasonable outputs, however, its failure rate is still much higher than ours. DreamBooth [6] however generates almost perfect images except for the 'blurry' text on the berry bowl. Through the comparison, we can observe remarkable improvement in the output image quality.

**Skillset.** We provide `SuTI`'s generation to showcase its ability in re-contextualization, novel view synthesis, art rendition, property modification, and accessorization. We demonstrate these different skills in the Appendix Figure 8. In the first row, we show that `SuTI` is able to synthesize the subjects with different art styles. In the second row, we show that `SuTI` is able to synthesize the different view angles of the given subject. In the third row, we show that `SuTI` can modify subjects' facial expressions like 'sad', 'screaming', etc. In the fourth row, we show that `SuTI` can alter the color of a given toy. In the last two rows, we show that `SuTI` can add different accessories (hats, clothes, etc) to the given subjects. Further, we found that `SuTI` can even compose two skills together to perform highly complex image generation. As depicted in Figure 5, we show that `SuTI` can combine re-contextualization with editing/accessorization/stylization to generate high-quality images.

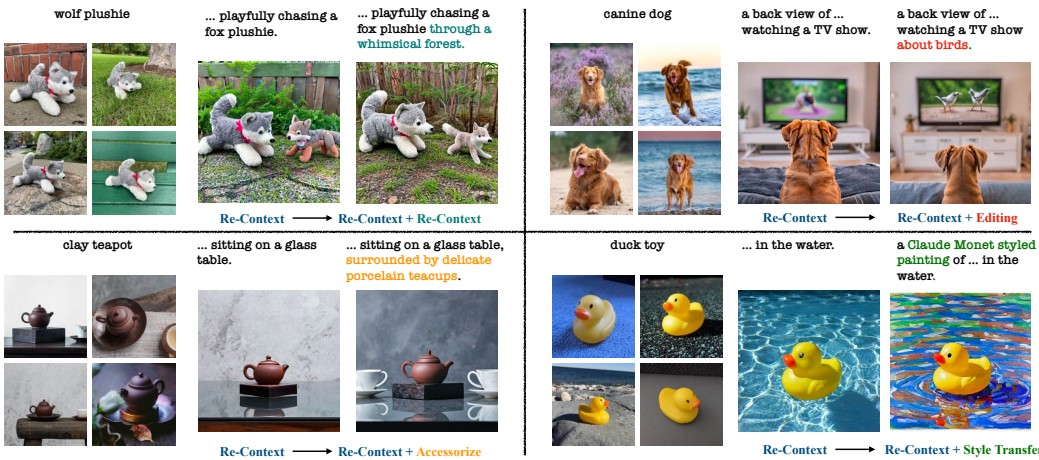

Figure 5: `SuTI` not only re-contextualizes subjects but also composes multiple transformation, all in-context.

## 5.3 Model Analysis and Ablation Study

We further conducted a set of ablation studies to show factors that impact the performance of `SuTI`.

**Impact of # Demonstrations.** Figure 6 presents the `SuTI`'s in-context generation with respect to an increasing number of subject-specific image examples. Interestingly, we observe a transition in the model's behavior as the number of in-context examples increases. When $\mathbb{C}_s = \varnothing$, `SuTI` generates images using it prior to the text, similar to traditional text-to-image generation models such as Imagen [1]. When $|\mathbb{C}_s| = 1$, `SuTI` behaves similarly to an image editing model, attempting to edit the generation while preserving the foreground subject, and avoiding sophisticated transformation. When $|\mathbb{C}_s| = 5$, `SuTI` unlocks the capability of rendering novel pose and shape of the demonstrated subject naturally in the targeted scene. In addition, we also observe that a bigger $|\mathbb{C}_s|$ would result in a more robust generation of high text and subject alignment, and better photorealism. We also performed a human evaluation on the `SuTI`'s generation with respect to different numbers of demonstrations and visualizes the results in Figure 6 (right). It shows that as the number of demonstrations increases, the human evaluation score first increases drastically and then gradually converges.

**Quality of the expert dataset matters.** We found that the Delta CLIP score is critical to ensure the quality of synthesized target images. Such a filtering mechanism is highly influential in terms of `SuTI`'s final performance. We evaluated several versions to increase the $\Delta$ threshold from None $\rightarrow$ $0.0 \rightarrow \cdots \rightarrow 0.025$, we observe that the human evaluation can steadily increase from 0.54 to 0.82.

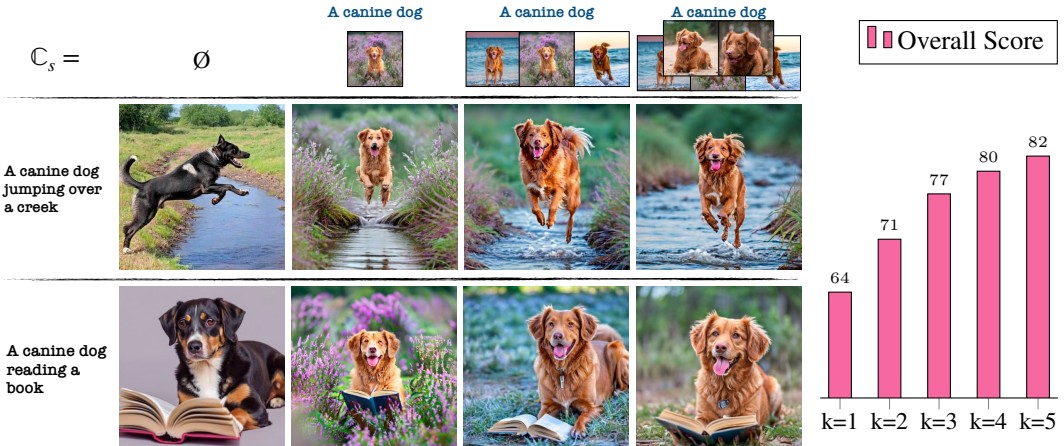

Figure 6: (Left) In-context generation by `SuTI` model, with an increasing # of demonstrations. (Right) Human evaluation score with with respect to the increasing % of demonstrations.

Without such intensive filtering, the model's overall human score can go to a very low level (54%). With an increasing $\Delta$, although the size of the dataset $G$ keeps decreasing from 1.8M to around 500K, the model's generation quality keeps improving until saturation. The empirical study indicates that $\Delta = 0.02$ strikes a good balance between the quality and quantity of the expert-generated dataset $G$.

| Methods | Inference Time | Subject ↑ | Text ↑ | Photorealism ↑ | Overall |
|---|---|---|---|---|---|
| DreamBooth | 10 secs | 0.88 | 0.82 | 0.98 | 0.77 |
| SuTI | 20 secs | 0.90 | 0.90 | 0.92 | 0.82 |
| Dream-SuTI | 15 secs | **0.92** | **0.92** | **0.94** | **0.87** |

Table 3: Quantitative Human evaluation of the `Dream-SuTI` model on DreamBench-v2.

**Further fine-tuning SuTI improves generation quality** We note that our model is not exclusive to methods that requires further fine-tuning, such as DreamBooth [6]. Instead, SuTI can be combined with DreamBooth [6] naturally to achieve better quality subject-driven generation (dubbed as `Dream-SuTI`). Specifically, given $K$ reference images regarding a subject, we can randomly feed one image as the condition and use another differently sampled image as the target output. Through fine-tuning the SuTI model for 500 steps (without any auxiliary loss), the Dream-SuTI model can generate aligned and faithful results for a given subject. Table 3 shows a comparison of the Dream-SuTI, against SuTI and DreamBooth, suggesting that `Dream-SuTI` further improves the generation quality. Particularly, it improves the overall score from 0.82 to 0.87, yielding a 5% improvement over SuTI, and 10% improvement over DreamBooth. Since the fine-tuned `Dream-SuTI` model already trained on all subject images, only one subject image is needed to present during the inference time, which can further reduce the inference cost.

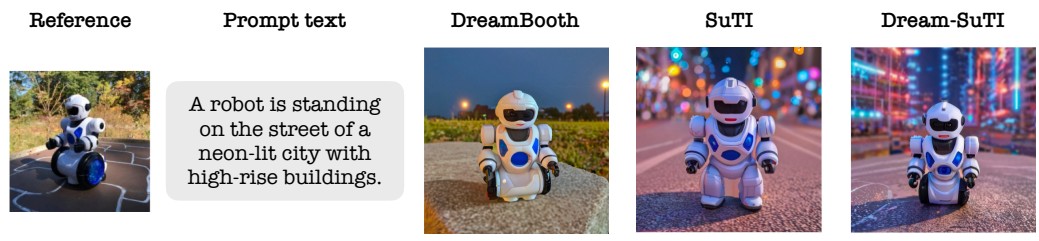

Figure 7: Comaprison between DreamBooth, SuTI and Dream-SuTI.

To gain better understanding of the quality, we show an example in Figure 7, where we pick a failure example from SuTI to investigate whether `Dream-SuTI` improves it. We observe that the DreamBooth does not have strong text alignment, while SuTI's subject lacks fidelity (the generated robot uses legs instead of wheels). With further fine-tuning on subject images, Dream-SuTI is able

to generate images not only faithful to the subject but also to the text description. However, we would like to note that such subject-driven fine-tuned model share the same drawback of a typical Dreambooth model, which can no longer generalize well to a general distribution objects and hence requiring a copy of model parameter per subject.

# 6 Related Work

**Text-Guided Image Editing** With the surge of diffusion-based models, [22, 8] have demonstrated the possibilities to manipulate given image without human intervention. Blended-Diffusion [23] and SDEdit [24] propose to blend the noise with the input image to guide the image synthesize process to maintain the original layout. Text2Live [25] generates an edit layer on top of the original image/video input. Prompt-to-Prompt [26] and Null-Text Inversion [12] aims at manipulating the attention map in the diffusion model to maintain the layout of the image while changing certain subjects. Imagic [7] propose an optimization based to achieve significant progress in manipulating visual details in a given image. InstructPix2Pix [11] propose to distill image editing training pairs synthesized from Prompt-to-Prompt into a single diffusion model to perform instruction-driven image editing. Our method resembles InstructPix2Pix in a sense that we are training the model on expert-generated images. However, our synthesized data is generated generated by fine-tuned experts, which are mostly natural images. In contrast, the images from InstructPix2Pix are synthetic images. In the experiment section, we comprehensively compare with these existing models to show the advantage of our model, especially on more challenging prompts.

**Subject-Driven Text-to-Image Generation** Subject-Driven Image Generation tackles a new challenge, where the model needs to understand the visual subject contained in the demonstrations to synthesize totally new scene. Several GAN-based models [27, 28] pioneered to work on personalizing the image generation model to a particular instance. Later on, DreamBooth [6] and Textual Inversion [8, 29] propose optimization-based approach to adapt image generation to a specific unseen subject. However, these two methods are time and space-consuming, which makes them unrealistic in real-world applications. Another line of work adopt retrieval-augmented architecture for subject-driven generation including KNN-Diffusion [30], Re-Imagen [9], however, these methods are trained with weakly-supervised data leading to much worse faithfulness. In this paper, we aim at developing an apprenticeship learning paradigm to train the image generation model with stronger supervision demonstrated by fine-tuned experts. As a result, SuTI can generate customized images about a specified subject, without requiring any test-time fine-tuning. There are some concurrent and related works [31, 32] focusing on specific visual domains such as human faces and / or animals. To our best knowledge, SuTI is the first subject-driven text-to-image generator that operates fully in-context, *generalizing* across various visual domains.

# 7 Conclusion

Our method SuTI has shown strong capabilities to generate personalized images instantly without test-time optimization. Our human evaluation indicates that SuTI is already better in the overall score than DreamBooth, however, we do identify a few weakness of our model: (1) SuTI's generations are less diverse than DreamBooth, and our model is less inclined to transform the subjects' poses or views in the new image. (2) SuTI is less faithful to the low-level visual details than DreamBooth, especially for more complex and often manufactured subjects such as 'robots' or 'rc cars' where the subjects contain highly sophisticated visual details that could be arbitrarily different from the examples inside the training dataset. In the future, we plan to investigate how to further improve these two aspects to make SuTI's generation more diverse and detail-preserving.

## Acknowledgement

We thank Boqing Gong, Kaifeng Chen for reviewing an early version of this paper in depth, with valuable comments and suggestions. We thank Neil Houlsby, Xiao Wang and also the PaLI-X team for providing early access to their Episodic WebLI data. We also thank Jason Baldbridge, Andrew Bunner, Nicole Brichtova for discussions and feedback on the project.

## Broader Impact

Subject-driven text-to-image generation has wide downstream applications, like adapting certain given subjects into different contexts. Previously, the process was mostly done manually by experts who are specialized in photo creation software. Such manual modification process is time-consuming. We hope that our model could shed light on how to automate such a process and save huge amount of labors and training. The current model is still highly immature, which can fall into several failure modes as demonstrated in the paper. For example, the model is still prone to certain priors presented in certain subject classes. Some low-level visual details in subjects are not perfectly preserved. However, it could still be used as an intermediate form to help accelerate the creation process. On the flip side, there are risks with such models including misinformation, abuse and bias. See the discussion of broader impacts in [1, 4] for more discussion.

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

# A Supplementary Material

## A.1 Dataset Construction

To validate the effectiveness, we provide an ablation study to show that higher precision is more important than recall in training the apprentice model. Particularly, when the threshold is set to a lower number (*e.g.*, 0.01 or 0.015), `SuTI` becomes less stable.

As our goal is to collect images of the same subject, we create an initial subject cluster by grouping all (image, alt-text) pairs that come from the same URL (∼45M clustrers), and filter the cluster with less than 3 instances (∼77.8% of the clusters). As a result, it leaves us with ∼10M image clusters. We then apply the pre-trained CLIP ViT-L14 model [33] to filter out 81.1% of clusters that has the average intra-cluster visual similarity between 0.82 and 0.98 to ensure the quality of clusters.

Though the mined clusters already contain (image, alt-text) information, the alt-text's noise level is too high. Therefore, we apply the state-of-the-art image captioning model [10] to generate descriptive text captions for every image of all image clusters, which forms the data triples of (image, alt-text, caption). However, current image captioning models tend to generate generic descriptions of the visual scene, which often occlude the detailed entity information about the subject. For example, generic captions like '`a pair of red shoes`' would greatly decrease the expert model's capability to preserve the subject's visual appearance. To increase the specificity of the visual captions, we propose to merge the alt-text, which normally contains specific meta information like brands, names, etc with the model-generated caption. For example, Given an alt-text of '`duggee talking puppet hey duggee chicco 12m`' and a caption of '`a toy on the table`', we aim to combine them as a more concrete caption: '`Hey duggee toy on the table`'. To achieve this, we prompt the pre-trained large language models [18] to read all (alt-text, caption) pairs inside each image cluster, and output a short descriptive text about the visual subject. These refined captions with the mined images are used as the image-text cluster $\mathbb{C}_s$ w.r.t subject $s$, which will be used to fine-tune the expert models.

## A.2 `SuTI` Skillset

We demonstrate the complete view of `SuTI`'s skillset in Figure 8, including styled subject generation, multi-view subject rendering, subject expression modification, subject colorization, and subject accessorization.

## A.3 Failure Examples

Figure 9 show some failure examples of `SuTI`. We show several types of failure modes: (1) the model has a strong prior about the subject and hallucinates the visual details based on its prior knowledge. For example, the generation model believes 'teapot' should contain a '`lift handle`'. (2) some artifacts from the demonstration images are being transferred to the generated images. For example, the '`bed`' from the demonstration is being brought to the generation, (3) the subject's visual appearance is being modified through, mostly influenced by the context, like the '`candle`' contains non-existing artifacts when contextualized in the '`toilet`'. These three failure modes constitute most of the generation errors. (4) The models are not particularly good at handling compositional prompts like the '`bear plushie`' and '`sunglasses`' example. In the future, we plan to work on how to improve these aspects.

## A.4 More Qualitative Examples

We demonstrate more examples from DreamBench-v2 in the following figures:

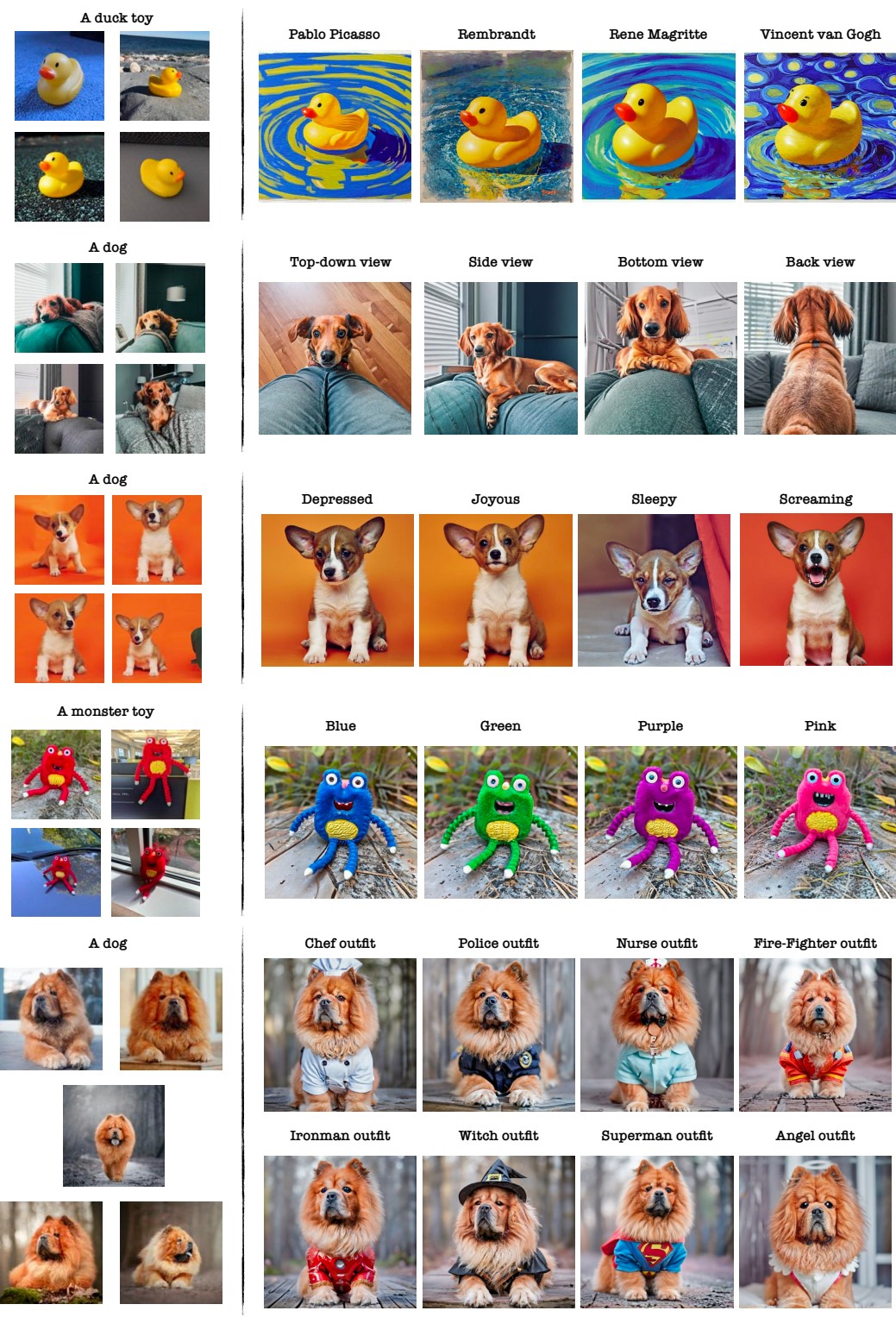

Figure 8: SuTI's in-context generation that demonstrates its skill set. Results generated from *a single model*. First row: art rendition of the subject. Second row: multi-view synthesis of the subject. Third row: modifying expression for the subject. Fourth row: editing the color of the subject. Fifth row: adding accessories to the subject. Subject (image, text) and editing key words are annotated, with detailed template in the Appendix.

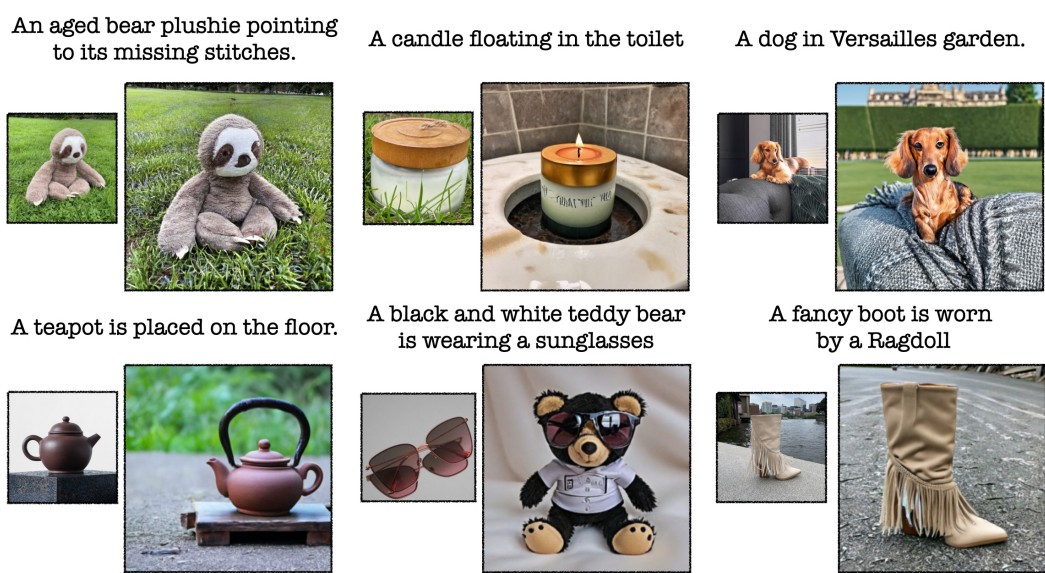

Figure 9: SuTI's failure examples on DreamBench-v2.

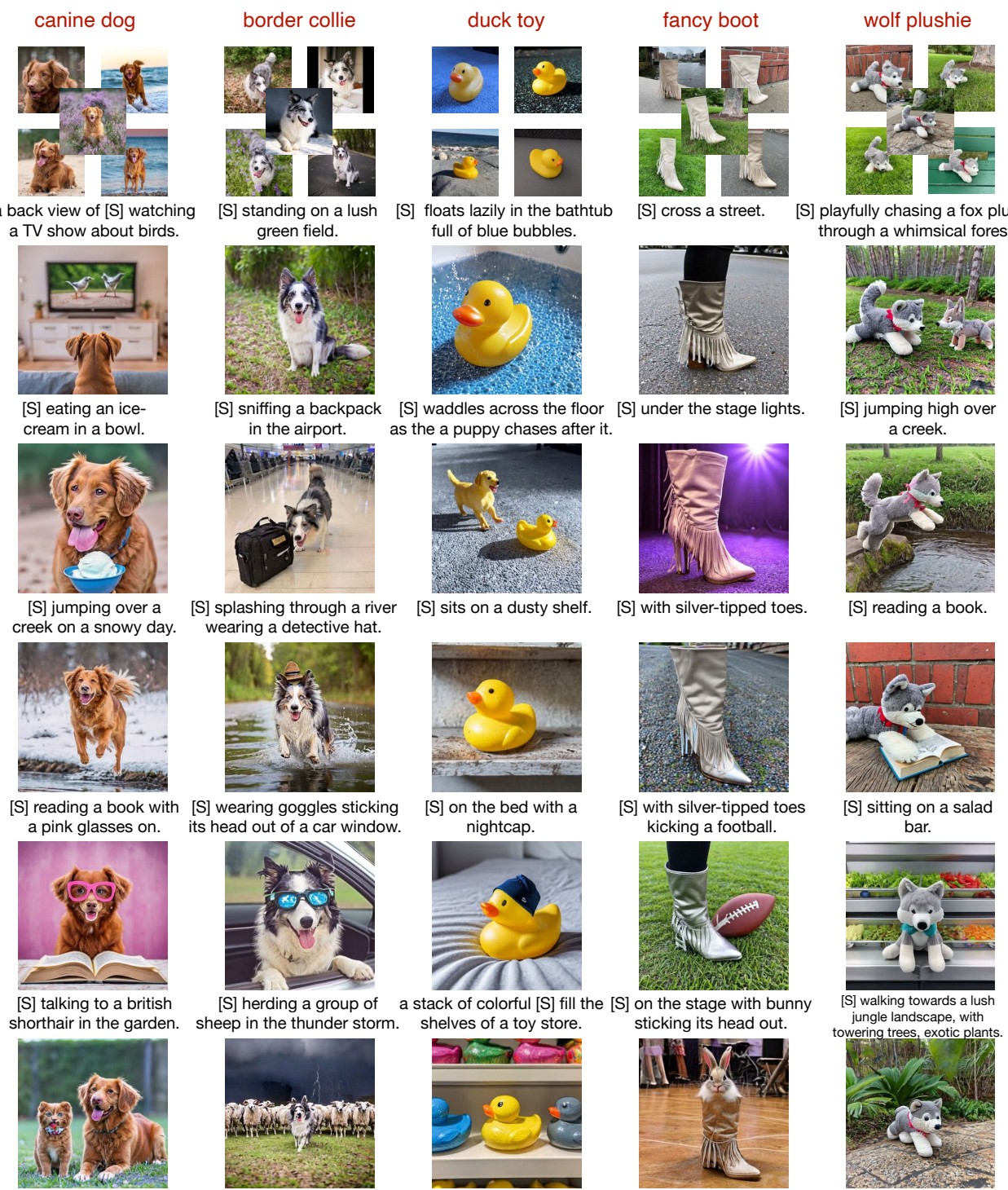

Figure 10: Visualization of SuTI's generation on the DreamBench-v2 (Part 1).

Figure 11: Visualization of SuTI's generation on the DreamBench-v2 (Part 2).

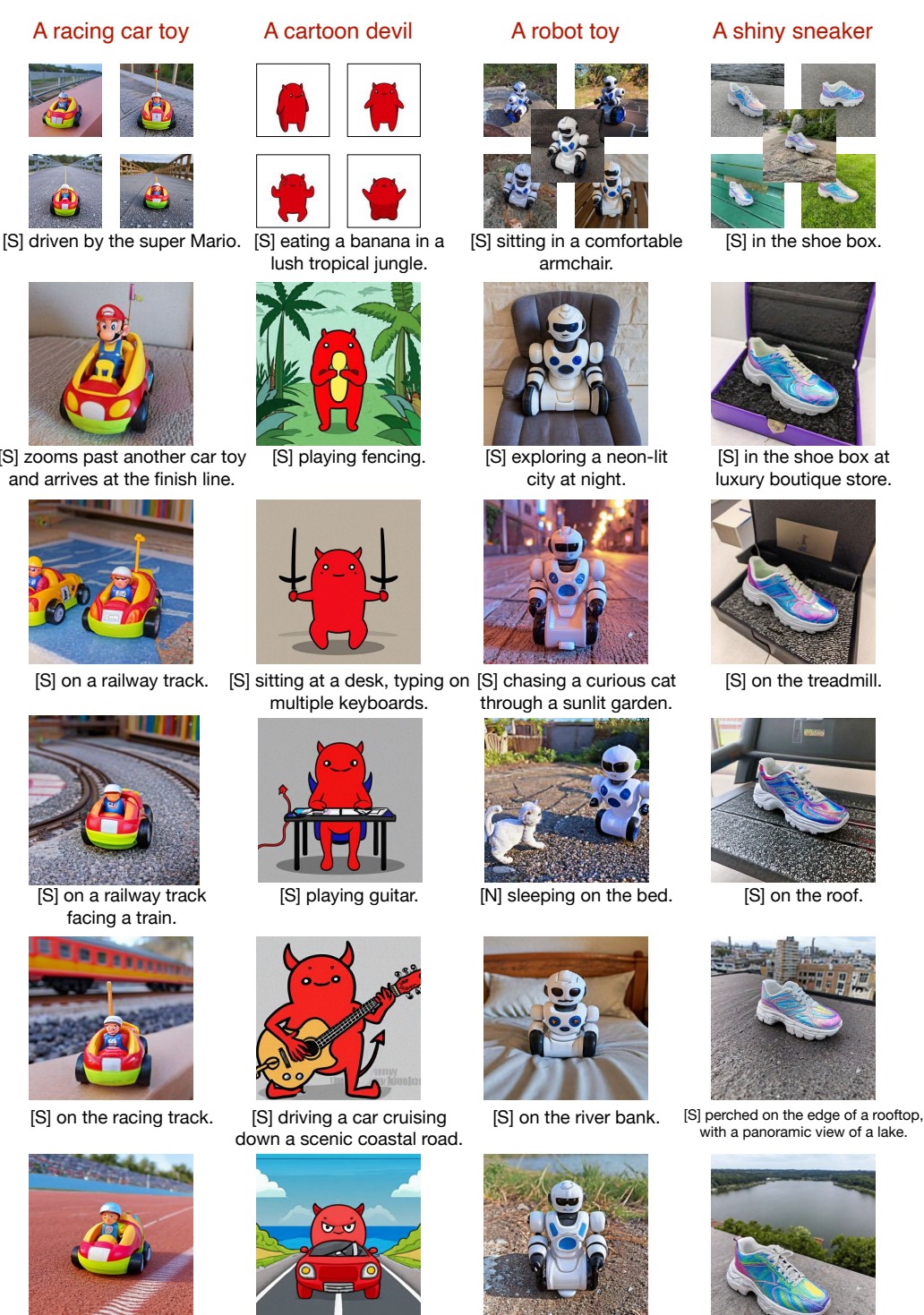

Figure 12: Visualization of SuTI's generation on the DreamBench-v2 (Part 3).

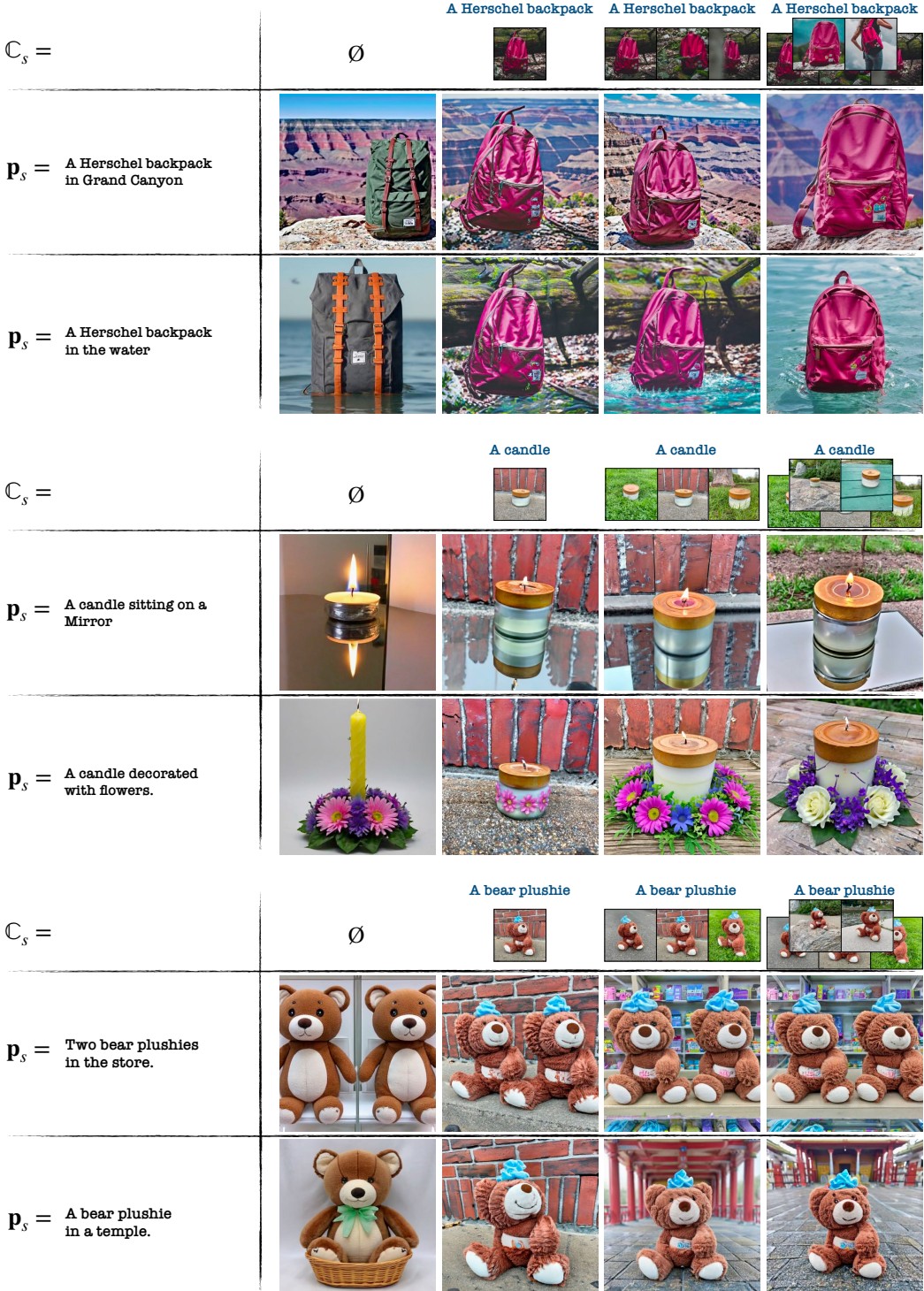

Figure 13: In-context generation by `SuTI` model, with an increasing # of demonstration (More examples).

