# A  Supplementary Material

## A.1  Extended Related Work

**Text-to-Image Generation** Previously, different GAN-based models [32, 33, 34, 35] have shown great progress in generating high-quality images. Recently, diffusion-based models models [36, 37, 1, 5, 38, 4] have gained unprecedented popularity to surpass the GAN-based models. These models have shown great progress in generating highly realistic images faithful to the given text control. The progress is mainly driven by diffusion model [14, 15] and auto-regressive backbone [3]. However, these models can only accept text prompt as the input, lacking control from other sources. For example, if we want to generate an image about our own dog or our own backpack in different scenes, it becomes challenging for the existing models [6]. Also, as suggested by [9], the existing generation models are highly biased towards generating frequent subjects while having difficulty generating less common visual entities. These challenges have spawned the new task of 'Subject-Drive Text-to-Image Generation', which is the core task of our paper aims to solve.

## A.2  Dataset Construction

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

## A.6 More Examples

We demonstrate more examples from DreamBench-v2 in the following:

| canine dog | border collie | duck toy | fancy boot | wolf plushie |
|---|---|---|---|---|
| a back view of [S] watching a TV show about birds. | [S] standing on a lush green field. | [S] floats lazily in the bathtub full of blue bubbles. | [S] cross a street. | [S] playfully chasing a fox plushie through a whimsical forest. |
| [S] eating an ice-cream in a bowl. | [S] sniffing a backpack in the airport. | [S] waddles across the floor as the a puppy chases after it. | [S] under the stage lights. | [S] jumping high over a creek. |
| [S] jumping over a creek on a snowy day. | [S] splashing through a river wearing a detective hat. | [S] sits on a dusty shelf. | [S] with silver-tipped toes. | [S] reading a book. |
| [S] reading a book with a pink glasses on. | [S] wearing goggles sticking its head out of a car window. | [S] on the bed with a nightcap. | [S] with silver-tipped toes kicking a football. | [S] sitting on a salad bar. |
| [S] talking to a british shorthair in the garden. | [S] herding a group of sheep in the thunder storm. | a stack of colorful [S] fill the shelves of a toy store. | [S] on the stage with bunny sticking its head out. | [S] walking towards a lush jungle landscape, with towering trees, exotic plants. |

Figure 9: Visualization of SuTI's generation on the DreamBench-v2 (Part 1).

A grey sloth plushie     A red monster toy     Pink sunglasses     A poop emoji toy     A clay teapot

[S] climbing a tree.    [S] sitting on a wing chair.    [S] hang on the wall.    [S] on a clock tower.    [S] on a glass table.

[S] dangles lazily from a backpack.    [S] sitting on a wing chair with a teddy bear.    [S] on a wooden deck overlooking a lake.    [S] under the Tokyo tower.    [S] pouring steaming hot water into a teacup.

[S] reading a paper.    [S] having sushi.    [S] sitting on a river bank facing skycrapers.    [S] talking to a red heart emoji toy    [S] sitting on a glass table, surrounded by delicate porcelain teacups.

[S] wearing a T-shirt.    [S] on the book cover.    [S] in a yellow sunglass case.    [S] wearing a big nose funny glasses.    [S] on the wooden table, together with a salmon sushi.

An aged [S]    [S] flying a kite in the desert.    [S] in the microwave oven.    [S] in a hot air ballon in the sunset.    [S] on the floor, surrounded by scattered tea leaves.

Figure 10: Visualization of SuTI's generation on the DreamBench-v2 (Part 2).

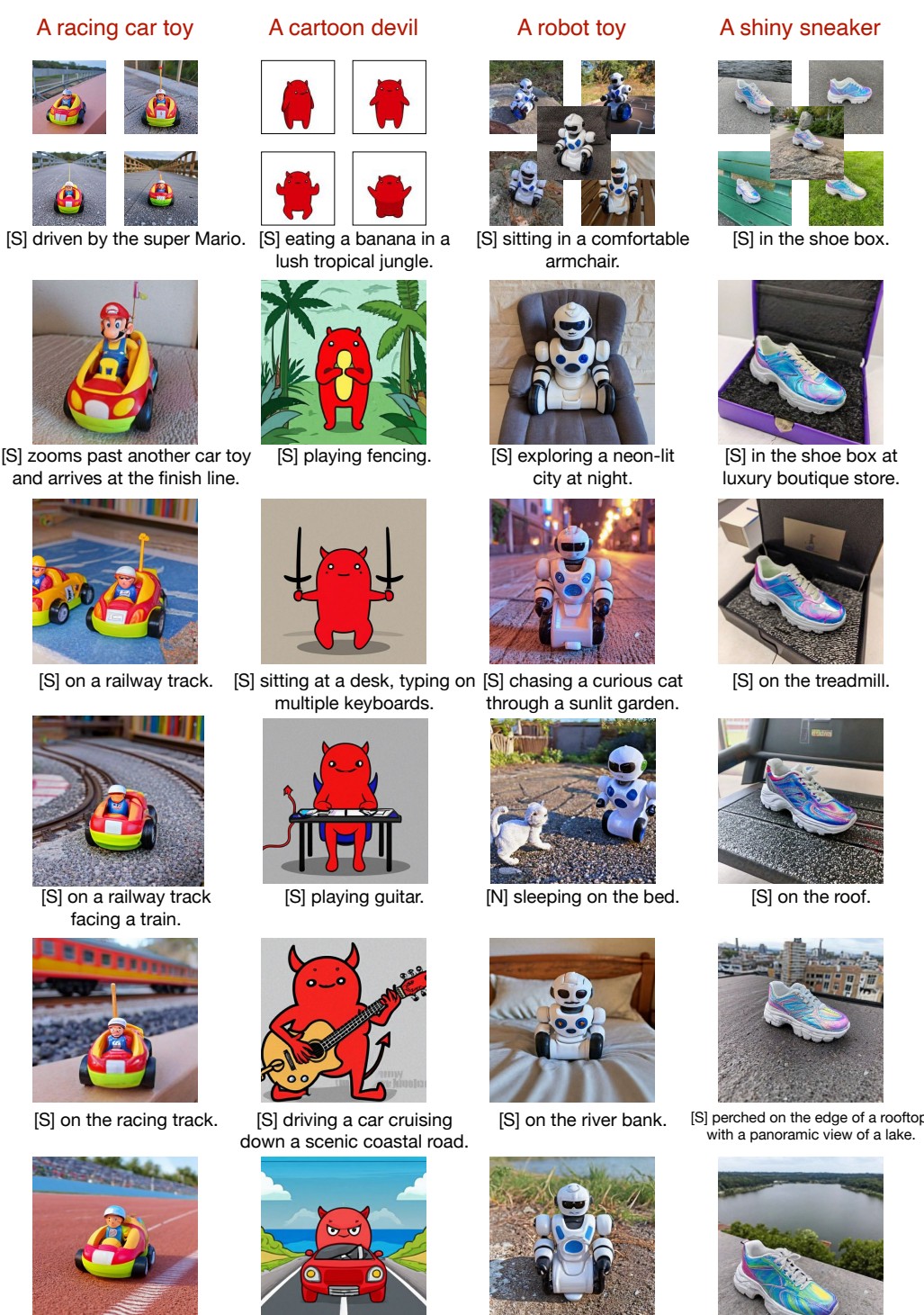

Figure 11: Visualization of SuTI's generation on the DreamBench-v2 (Part 3).

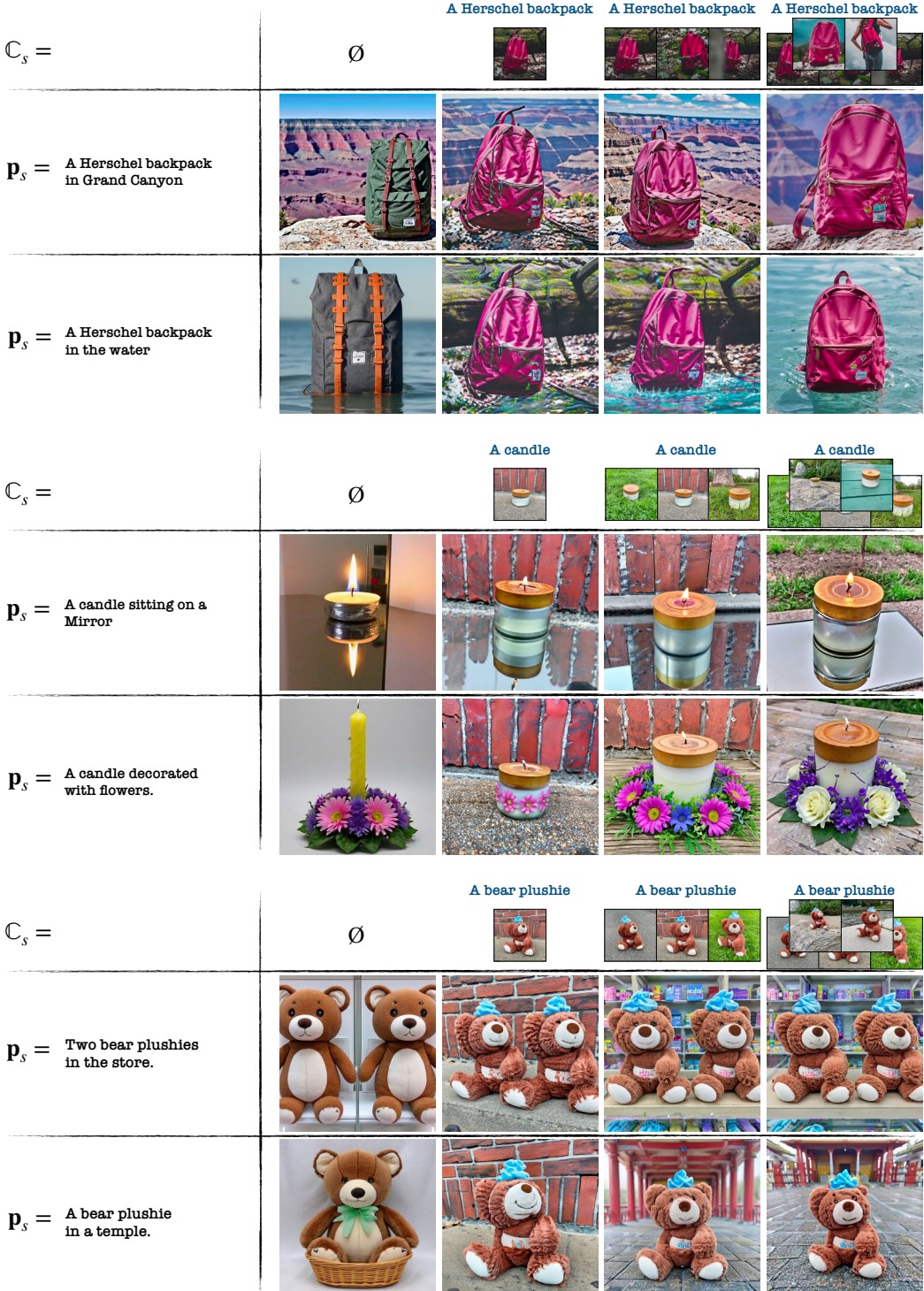

Figure 12: In-context generation by `SuTI` model, with an increasing # of demonstration (More examples).