# OpenReview forum: "Subject-driven Text-to-Image Generation via Apprenticeship Learning"
_NeurIPS.cc/2023/Conference — NeurIPS 2023 poster_

### Official Review · Reviewer_HXFy · 2023-07-05

**Soundness:** 3 good
**Presentation:** 3 good
**Contribution:** 2 fair
**Rating:** 6
**Confidence:** 5

**Summary:**

This paper proposes a method for subject-driven text-to-image generation, where a model is tasked to generate novel renditions of a subject given a few images of that subject. Different from previous fine-tuning approaches, this paper trains a model that conditions its generation on the given subject images. To train such a model, this paper first trains a large amount of subject-specific fine-tuned models, and use these fine-tuned models to generate new training data for knowledge distillation. The distilled model achieves good qualitative performance.


**Strengths:**

- It is nice that the apprenticeship model does not require finetuning on new subjects. It is an interesting idea to directly use subject images as the conditional input for generation.
- The paper is mostly well-written.
- The learned model achieves good qualitative results.

**Weaknesses:**

- The training data covers a wide range of subjects. Therefore, it is hard to tell if the apprenticeship model learns to generalize to new subject or not. Have the authors performed de-duplication to ensure that the subjects used for evaluation do not appear in the training set of the expert models?
- The proposed method requires training of 2M expert models, where each expert is a 2.1B Imagen model. This is extremely expensive both computational-wise and storage-wise.
- The inference speed of the apprenticeship model seems to be slow, as each demonstration sample needs to pass through the Imagen model. Methods such as DreamBooth do not incur additional inference cost after the finetuning.
- It would be nice to see some ablation on the importance of the expert model. For each subject cluster, how many synthetic images are used to train the apprenticeship model? What if only the real images are used to train the apprenticeship model?
- It would be better if the paper could give a more detailed illustration of the apprenticeship model's architecture, instead of referring to the ReImagen paper.


**Questions:**

- Is it possible to finetune the apprenticeship model on new subjects?

**Limitations:**

Yes the authors have addressed the limitations.

---

> ### Author Rebuttal · Authors · 2023-08-04
>
> Comment #1 “The training data covers a wide range of subjects. Therefore, it is hard to tell if the apprenticeship model learns to generalize to new subjects or not. Have the authors performed de-duplication to ensure that the subjects used for evaluation do not appear in the training set of the expert models?”
>
> We investigated whether the validated concepts appear in the training dataset. We use CLIP score to retrieve nearest cluster images for each of the 30 concepts in DreamBenchv2. We manually check whether these concepts appear in the training set. We found that very few subjects have similar variants in the training set, and these are mostly  “dogs” and “cats”. Most other subjects like “Robot”, “Vase” are very unique and we can’t find even modestly similar ones.
>
> Comment #2 “The proposed method requires training of 2M expert models, where each expert is a 2.1B Imagen model. This is extremely expensive both computational-wise and storage-wise.”
>
> Our method is computationally expensive, but since we don’t store any DreamBooth checkpoints, our storage consumption is very low, i.e. only a few hundred Gigabytes.
>
> Comment #3 “The inference speed of the apprenticeship model seems to be slow, as each demonstration sample needs to pass through the Imagen model. Methods such as DreamBooth do not incur additional inference cost after the finetuning.”
>
> There are two solutions to address this issue:
> (1) SuTI is highly compatible with DreamBooth. We can use the tuned SuTI just like DreamBooth by setting #demonstration=0 during train/inference time, which would lead to the exact same performance. If we want to gain better results, we could provide a single demonstration to the subject-tuned-SuTI, which will yield much better subject fidelity than DreamBooth with only marginal inference overhead (15 secs vs 10 secs).
> (2) We can use distillation in https://arxiv.org/abs/2210.03142 to shorten the diffusion steps by 10x or even more, which can decrease the inference time significantly.
>
> Comment #4 “It would be nice to see some ablation on the importance of the expert model. For each subject cluster, how many synthetic images are used to train the apprenticeship model? What if only the real images are used to train the apprenticeship model?”
>
> Thanks for the suggestion. We will put more ablation studies to the paper revision.
> 1. We have experiments to show how the model performance changes w.r.t the number of clusters, and what’s the minimum number of clusters needed to train SuTI.
> 2. We also have conducted experiments to see whether we could just use the clustered images (without DreamBooth expert) to train SuTi. Specifically, we use k-1 images as exemplars and the k-th as the target. However, due to the fact that the clustered images are so similar or near duplicate to each other, this training will guide the model to only copy-paste from demonstration. That’s why we will need more diverse outputs from DreamBooth to discourage the copy-paste behavior. We will add more analysis about this discovery into Appendix.
>
> Comment #5: “It would be better if the paper could give a more detailed illustration of the apprenticeship model's architecture, instead of referring to the ReImagen paper.”
>
> Thanks for the suggestion. We will add detailed model architecture to the Appendix in the revision.
>
> Comment #6 “Is it possible to finetune the apprenticeship model on new subjects?”
>
> Yes! We found that fine-tuning on SuTI is better than fine-tuning on the original Imagen model by a significant margin, particularly in terms of the subject fidelity. We plan to include some of these results in the revision.

---

> > ### Comment · Reviewer_HXFy · 2023-08-13
> >
> > Thanks the authors for their response.  Most of my concerns have been addressed, thus I will raise my score. Despite some weaknesses of this paper (such as its computational cost and close-sourced model), I still lean towards acceptance.

---

### Official Review · Reviewer_9TJ3 · 2023-07-06

**Soundness:** 3 good
**Presentation:** 3 good
**Contribution:** 3 good
**Rating:** 7
**Confidence:** 4

**Summary:**

The paper presents a novel subject-driven text-to-image generator named SuTI. This model leverages in-context learning as opposed to subject-specific fine-tuning. SuTI is built upon the principles of apprenticeship learning and is capable of generating high-quality, customized, subject-specific images. Remarkably, it achieves this at a speed that is 20 times faster than optimization-based methods.

SuTI has demonstrated superior performance over existing models on benchmark tests such as DreamBench and DreamBench-v2. The paper highlights the recent advancements in text-to-image generation models, which have shown significant progress in generating highly realistic, accurate, and diverse images from given text prompts.

**Strengths:**

The paper exhibits several strengths across the dimensions of originality, quality, clarity, and significance:

1. Originality: The paper introduces SuTI, a novel subject-driven text-to-image generator that uses in-context learning instead of subject-specific fine-tuning. This approach is original and innovative, as it deviates from the conventional optimization-based methods, offering a faster and more efficient solution.

2. Quality: The quality of the paper is evident in the rigorous testing and validation of the SuTI model. The model has been benchmarked against existing models on DreamBench and DreamBench-v2, where it has shown superior performance. This demonstrates the robustness and reliability of the model.

3. Clarity: The paper is well-structured and clear in its presentation of the SuTI model. It provides a comprehensive explanation of the model's workings, its applications, and its performance in various tests. The use of visual aids and examples further enhances the clarity of the paper.

4. Significance: The significance of the paper lies in its contribution to the field of text-to-image generation. By introducing a faster and more efficient model, the paper pushes the boundaries of what is currently possible in this field. This could have far-reaching implications for a variety of applications, including content creation, design, and more.

**Weaknesses:**

One weakness of the paper is the lack of discussion around the cost and complexity of constructing the training dataset. The process of creating a comprehensive and diverse dataset for training a model like SuTI can be a significant undertaking, both in terms of time and resources.

The paper does not delve into the specifics of this process, leaving readers without a clear understanding of the potential challenges and costs associated with data collection and preparation. This lack of transparency may make it difficult for others to replicate the study or apply the model in different contexts.

**Questions:**

1. Dataset Construction: Could you provide more details about the process of constructing the training dataset for SuTI? Specifically, how much time and resources were required to create the dataset? Were there any significant challenges encountered during this process?

2. Model Scalability: How scalable is the SuTI model with respect to the size and diversity of the training dataset? If the dataset were to be expanded or updated with new subjects or scenarios, would this significantly impact the model's performance or the resources required for training?



**Limitations:**

The authors have adequately addressed the limitations and potential negative societal impact of their work.

---

> ### Author Rebuttal · Authors · 2023-08-04
>
> Comment #1 “Dataset Construction: Could you provide more details about the process of constructing the training dataset for SuTI? Specifically, how much time and resources were required to create the dataset?”
>
> Since we can parallelize the DreamBooth training and generation, we set up 100 instances of TPU v4-8  (128G with 4 chips each). Each instance will take one “subject cluster” and then train one DreamBooth for 500 steps to generate the outputs. Once the output is generated, the model will reset to original weight to move on to the next “subject cluster” without saving any checkpoint. The whole dataset construction takes roughly 100 TPU v4-8 for two weeks.
> We don’t need to store the DreamBooth checkpoint, so the storage cost is negligible.
>
> Comment #2 “Were there any significant challenges encountered during this process?”
>
> This is a very insightful question. Yes, we did encounter some challenges:
> (1) The memory consumption of the original DreamBooth recipe is very high due to the usage of Adam optimizer, which already reaches the memory ceiling of TPU v4-8. Somehow the memory reset will cause an OOM issue. We replaced Adam with Adafactor to resolve this issue, which decreases the memory consumption significantly to enable our pipeline.
> (2) The other difficulty is how to filter the DreamBooth failure outputs. We tested different variants and conducted lots of human study to decide the CLIP-delta threshold. But still, this threshold is not perfectn with lots of false positives and false negatives.  The community is  still in dire need for a better automatic metric to evaluate subject fidelity.
>
> Comment #3 “Model Scalability: How scalable is the SuTI model with respect to the size and diversity of the training dataset? If the dataset were to be expanded or updated with new subjects or scenarios, would this significantly impact the model's performance or the resources required for training?”
>
> So far, we found the model to be quite scalable with respect to the number of subjects and skill sets we added to the training dataset. For now, we have 5-6 skill-sets and millions of subjects in the dataset. We are still exploring newer skillsets right now, like image editing, image inpainting, etc. One possible way to extend the model’s capacity is to use a small adapter layer to increase models’ capacity to handle more diverse inputs.

---

### Official Review · Reviewer_F8hk · 2023-07-06

**Soundness:** 3 good
**Presentation:** 3 good
**Contribution:** 3 good
**Rating:** 6
**Confidence:** 4

**Summary:**

This paper introduces SuTI for the subject-driven text-to-image (T2I) generation method. Numerous expert models are first trained on millions of image clusters collected from the internet, each focuses on a specific visual subject. A dataset is then created, consisting of concept images, target prompts, and corresponding target images. By training the apprentice model on this dataset, SuTI can generate high-quality and customized subject-specific images without the need for test time fine-tuning. Both the qualitative and quantitative experiments are performed with the baselines.

**Strengths:**

- The presented method is well-motivated and easy to understand.
- SuTI demonstrates fast generation and has a broad domain of applicability.
- The paper includes extensive experimental evaluations with impressive results.

**Weaknesses:**

Some details of the experiments require clarification:

- The hyperparameters used for training the baselines, such as the number of training iterations and learning rates, should be provided.
- More information is needed regarding the human evaluation, including the number of questions, the number of evaluated images, and the number of users involved.

**Questions:**

- More details about the experiments should be clarified.
- Can the proposed model perform multiple concept generation, which is also important in concept image generation?

**Limitations:**

yes

---

> ### Author Rebuttal · Authors · 2023-08-04
>
> Comment #1 “The hyperparameters used for training the baselines, such as the number of training iterations and learning rates, should be provided”:
>
> Thanks for the reminder. We mostly use the official colab from these papers and their default hyper-paramters for the baselines. We will add these details to the revision.
>
> Comment #2 “More information is needed regarding the human evaluation, including the number of questions, the number of evaluated images, and the number of users involved.”
>
> We will add our human evaluation guideline into the Appendix in the revision. The evaluation is done by several trained raters on the DreamBenchv2’s 220 images.
>
> Comment #3 “Can the proposed model perform multiple concept generation, which is also important in concept image generation?”
>
> We evaluated SuTI on 2-concept image generation. The model works well on some easy combinations like “dogs [D] besides vase [V]” or “cat [C] behind a bowl [B]”. But it would fail on more difficult combinations (with subject interaction) like “dog [D] wearing shoes [S]”, etc. We will put some of these results in the revision.

---

> > ### Author Response · Authors · 2023-08-18
> > **Follow-up**
> >
> > We would like to follow up on our rebuttal. If there are any additional outstanding concerns that you would like us to address, please let us know. Thank you and we look forward to your response.

---

> ### Comment · Reviewer_F8hk · 2023-08-18
> **Official Comment by Reviewer F8hk**
>
> Thanks to the authors for their efforts. I have read the rebuttal and the comments from the other reviewers, and most of my concerns have been addressed. I agree with Reviewer JnTg and look forward to the code being open source. I decided to raise my score from 5 to 6.

---

### Official Review · Reviewer_6um5 · 2023-07-06

**Soundness:** 3 good
**Presentation:** 4 excellent
**Contribution:** 3 good
**Rating:** 6
**Confidence:** 5

**Summary:**

The paper proposes an in-context learning method for model customization given personalized objects. The method first collects a large-scale dataset of custom concepts ensuring all images in each custom concept cluster are similar to each other and fine-tunes the model for each concept using Dreambooth. These expert models are then used to get the dataset to train the in-context learning method. It takes a few image text pair of the concept and a new text prompt to generate the image corresponding to the new prompt. The method is based on the Re-Imagen framework. To ensure high quality image text pair dataset from dreambooth models, CLIP based feature similarity threshold is applied.

**Strengths:**

The method is one of the first works on in-context learning for model customization. This prevents the time-consuming step of fine-tuning models given any new subject images. Both qualitative and quantitative results show that the method performs on par or better than existing zero-shot or fine-tuning methods.

The paper is well-written, easy to understand, and has extensive ablation experiments to validate the importance of different aspects of the method.


**Weaknesses:**

Why is there a need to train the Dreambooth expert models for training the final SuTI model? How does the performance change if the collected dataset itself is used directly to train the final model with N-1 image text pairs for in-context input and 1 sample as the new inference? It would be great to have an analysis regarding that.

Other objects included in the image often take the characteristics of the main subject, e.g., kite in Figure 10 2nd column last row or the british shorthair cat in Fig. 9 1st column last row in the appendix. Does adding a specific characterization in the text prompt for the other subject prevent that? Or is it overfitting in such scenarios? Does having more in-context demonstrations prevent that?

Can it be combined with other editing-based methods like SD-Edit or similar approaches to edit a specific region of the image or combine multiple specific subjects in the same image? E.g., to generate the canine dog of Figure 5 eating the cherry bowl of Figure 4. The editing example shown in Figure 5 changes the whole image instead of just the birds in the TV show.

Another point I would like to make is that it would have been great to also show the performance of SuTI with the SD backbone. This is not a weakness per se, and it doesn't affect the final rating. It's excellent work, but it would be helpful to assess the method's performance on such open-source models. Some of the baseline methods shown in the paper are already with the SD backbone.


**Questions:**

Minor questions and comments:

1. Does having a variable number of demonstrations instead of 3 have any effect on the final model performance?

2. It would be nice to include some sample descriptive text predicted by the language model for various subjects. Is it usually a class word corresponding to the subject or includes more details regarding the subject?

3. Is there any reason for low photorealism score of SuTI compared to DreamBooth?

4. Line 94. "to to" typo

5. Line 116. "models" -> "model"

6. Line 209. Probably a missing word in the sentence. "with largest space consumption"

7. Line 280. "generated generated" repeated words.

8. Line 286. “GAN=based” -> “Gan-based”

9. Line 306. "subjects are contain" -> "subjects contain"

10. Appendix Line 429. "Subject-Drive" -> "Subject-Driven"

11. Appendix Line 430. "task of our paper aims to solve" -> "task our paper aims to solve"



**Limitations:**

Yes.

---

> ### Author Rebuttal · Authors · 2023-08-04
>
> Thanks a lot for the highly detailed and constructive feedback!
>
> Comment #1  “​​Why is there a need to train the Dreambooth expert models for training the final SuTI model? How does the performance change if the collected dataset itself is used directly to train the final model with N-1 image text pairs for in-context input and 1 sample as the new inference? It would be great to have an analysis regarding that.”
>
> This is exactly our initial experiments to build SuTI. However, as the images in the same clusters are too similar, often near duplicates (particularly for rigid body objects), so after training, the model falls into a local optimum of copy-pasting the reference image without referring to the text instruction at all. We spent weeks of efforts to increase the diversity within the image cluster to discourage this behavior, however, the issue was not fully resolved.
>
> Therefore, we took the hard route of using DreamBooth + LLM to generate diverse subject images under different context/visual scenes to generate highly different outputs. This proves to be extremely effective in overcoming the copy-paste behavior. We will add more discussion about these failed efforts in the revision.
>
> Comment #2 “Other objects included in the image often take the characteristics of the main subject, e.g., kite in Figure 10 2nd column last row or the british shorthair cat in Fig. 9 1st column last row in the appendix. Does adding a specific characterization in the text prompt for the other subject prevent that? Or is it overfitting in such scenarios? Does having more in-context demonstrations prevent that?”
>
> This is a very common issue from the base text-to-image diffusion model, which sometimes does poorly on understanding language compositionality. We have some empirical evidence to support: (1) “adding more specific characterization” would generally help in some cases, e.g., the kite case can be fixed, but the “british shorthair” won’t be fixed. (2) “adding more in-context examples” does not seem to help much in this case. “More examples” can increase the identity preservation, but does not fully address the compositionality issue. (3) this is not necessarily “overfitting”, rather this compositionality issue is inherited from the original text-to-image generation model (see https://arxiv.org/abs/2212.05032).
>
> Comment #3  “Can it be combined with other editing-based methods like SD-Edit or similar approaches to edit a specific region of the image or combine multiple specific subjects in the same image? E.g., to generate the canine dog of Figure 5 eating the cherry bowl of Figure 4. The editing example shown in Figure 5 changes the whole image instead of just the birds in the TV show.”
> “Combine with Region-based in-painting”: Yes, it can definitely be combined with SD-Edit/DiffEdit to only in-paint a specific area of the image. We have some results that we haven't put it in the paper yet but will dosoin the revision.
> “Multiple Subject Composition”: We have tested this composition ability and found that SuTI does generalize to two subjects in some easy cases. But the failure rate is still pretty high, therefore, we did not include this part in the paper. Moving forward, multi-subject image generation will definitely be a priority.
>
> Comment #4 “Another point I would like to make is that it would have been great to also show the performance of SuTI with the SD backbone. This is not a weakness per se, and it doesn't affect the final rating. It's excellent work, but it would be helpful to assess the method's performance on such open-source models. Some of the baseline methods shown in the paper are already with the SD backbone.”
>
> We definitely want to make SuTI accessible to the general public. Since the submission, we have made some good progress toward this direction. Particularly, we are actively investigating the possibility of releasing the SuTI as a model-based model API service.
>
> Comment #5 Response to other individual questions.
>
> Q1: Yes, normally the more the better. But the training will be slower and memory consumption will be higher. We pick 3 as a trade-off between memory consumption vs quality.
> Q2: Sure, we will add more examples in the Appendix. The data contains both: (1) general class word corresponding like “shoes”, (2) more specific subject like “Nike Air … Shoes”. We hope this two modes can help the model ground better.
> Q3: There are two potential reasons: (1) SuTI fits the output of DreamBooth output, which has a slightly different distribution than natural image, especially under large classifier-free guidance. The model might overfit to this new distribution, (2) SuTI only has 2B parameters, while DreamBooth has 2B parameters per subject, which leads to different model capacity. This could cause the photorealism to be different.
> Q4-Q11: We will fix the typos.

---

> > ### Comment · Reviewer_6um5 · 2023-08-18
> > **Thanks for the response**
> >
> > The rebuttal addresses most of my concerns. Looking forward to the open-source model/API as well. Thanks.
> > I will keep my current rating.

---

### Official Review · Reviewer_JnTg · 2023-07-17

**Soundness:** 2 fair
**Presentation:** 4 excellent
**Contribution:** 3 good
**Rating:** 5
**Confidence:** 5

**Summary:**

In this paper, the authors propose a new model that can perform subject-driven text-to-image generation. Instead of fine tuning a leerte pretrained model on each subject, the authors use apprentice learning to first construct a virtual dataset from a large number of teacher models, each specific to a kind of subjects, and then have the student model learn from the constructed dataset. The authors have also shown strong qualitative and quantitative results and ablation study.

**Strengths:**

The paper is very well written and easy to follow. The results seem very promising. The paper tackles an interesting problem that is relevant to the applications.

**Weaknesses:**

1. This paper could have been impactful if the authors plan to open source or provide a way to reproduce the results. However, from the checklist the authors indicate that they have no plan to open source the model. Given that this model is trained with hundreds of TPUs, unless the authors provide the pretrained model or explicit instructions on how to reproduce the results in a non-cost prohibitive way, I don’t really see a way for the peer researchers to verify the results, nor do I see any real benefits for the practitioners from this paper.
2. The authors performed human evaluations. However, details about the instructions given to human evaluators and the compensation are not provided.

**Questions:**

Please address the weakness mentioned above. I am happy to change my score if problems related to my concerns are resolved.
How long does it take to train the apprentice model?

**Limitations:**

There is no discussion of the details about the human evaluation (I.e. the instructions given to human and the compensations).

---

> ### Author Rebuttal · Authors · 2023-08-04
>
> Comment #1 “This paper could have been impactful if the authors plan to open source or provide a way to reproduce the results. However, from the checklist the authors indicate that they have no plan to open source the model.”
>
> We are in full agreement that it would be far better for the community to release the code and/or model, and we are actively working towards doing that.  Although we cannot make guarantees at this point, we do plan to make the model or API publicly available to researchers before paper publication.
>
> When the paper was initially submitted, we had not yet established a concrete plan for open sourcing, so we were conservative and marked 'no' in the checklist to avoid potential miscommunication. Please note that according to the NeurIPS 2023 paper checklist guideline (https://neurips.cc/public/guides/PaperChecklist), particularly in the section 6 experiments, it is explicitly stated that “Papers cannot be rejected simply for not including code, unless this is central to the contribution (e.g., for a new open-source benchmark).
>
> Comment #2 “I don’t really see a way for the peer researchers to verify the results”:
>
> We respectfully disagree. First of all, we would release all of our test prompts, and generate results, which enables the researchers to do comparison. Secondly, we have included most key technical details in the paper, such that readers can run their experiments to reproduce the results. Finally, as stated above, we are actively working on releasing the model, to enable researchers to reproduce our results.
>
> Comment #3 “nor do I see any real benefits for the practitioners from this paper”:
>
> We believe that our paper provides some deep insight for the research community. For example, we found that fine-tuning on SuTI is better than fine-tuning on the original Imagen model. The subject fidelity improves quite a lot. We plan to put some of these results in the revision. The paper also provides an in-depth description of our novel methodology, which can potentially be replicated with smaller resources on public models if some constraints are relaxed. Moreover, as we mentioned, we are actively investigating the possibility of releasing the SuTI as a model-based model API service.
>
> Comment #4 “The authors performed human evaluations. However, details about the instructions given to human evaluators and the compensation are not provided”:
>
> Thanks for the comment. We have performed a training session to our human annotators to ensure their evaluation is calibrated, meanwhile the compensation to our annotator is paid hourly instead of per annotation, which incentivizes the annotators to spend more time for each annotation. We will put our annotation guideline and other details to the Appendix in the revision.
>
> In addition, we also like to mention that the automatic evaluation on DreamBench shows that SuTI outperforms other methods (best on 2 eval metric, and on par with Imagen based DreamBooth on 1 eval metric), which is aligned with the human evaluation.
>
> Comment #5 “How long does it take to train the apprentice model?”:
>
> The training time cost is less than 24 hours on 64 TPUs on Google Cloud Platform (actual duration depends on the version of TPU used. For your reference, training on TPU v4 takes ~10 hours).

---

> > ### Author Response · Authors · 2023-08-16
> > **About rebuttal**
> >
> > Thanks again for providing the detailed reviews! We are wondering whether you have read our rebuttal. If there is anything else we can discuss or clarify, please let us know. We are more than happy to discuss further during the author-reviewer discussion period.

---

> > ### Comment · Reviewer_JnTg · 2023-08-18
> > **Thank you for your response**
> >
> > Thank you for your response. Based on the prospective open source model, I will change my rating from rejection to acceptance. And I do want to comment that
> >
> > Comment #2 Since the model is extremely computationally expensive, even if the authors provide technical details in the paper, it is still very unlikely for general readers to reproduce the results without the open source model.
> >
> > Comment #3 I agree with the author rebuttal that the paper can provide some insight to the methodology.
> >
> > Thank you for your efforts to open source this model. Looking forward to its release.

---

> > > ### Author Response · Authors · 2023-08-18
> > > **Thank you for your response**
> > >
> > > Thanks a lot! We will try our best to release our model or API to the public.

---

### Author Rebuttal · Authors · 2023-08-09

We thank all the reviewers for their constructive feedback. Here we want to highlight a few things:

1. First of all, we are advocator of open research and strive to make everything publicly accessible. We plan to make the model or API publicly available to public before paper publication although we can't guarantee the exact date and time at this point.

2. Regarding the the combination of SuTI and DreamBooth, we conducted more experiments and show our results in the attached pdf.

3. Regarding the ablation study to only use the clustered images to train SuTI, i.e. without relying on the synthetic images by DreamBooth. We provide more evidence in the attached pdf.

---

### Comment · Area_Chair_v7aZ · 2023-08-19

Dear Reviewers,

The authors and I are eager to ascertain whether the author responses have effectively addressed your concerns. Due to the limited time allocated for the author-review discussion phase, we strongly encourage you to provide your direct feedback to the authors.

Thanks for your hard work.

Best regards,

AC

---

### Decision · Program_Chairs · 2023-09-21

**Decision:**

Accept (poster)

**Comment:**

This work proposes a subject-driven generation model without need for fine-tuning. Concerns are mainly on the training and inference computation cost, reproducibility, availability of the open-source model. After the rebuttal, concerns were mostly addressed and reviewers reached agreement upon acceptance. I agree with this recommendation and would strongly encourage authors to include reviewers’ suggestions during the revision, and make their model available for other researchers and practitioners, either through API or via open-sourcing.